# Acetylcholine esterase inhibitory activity of green synthesized nanosilver by naphthopyrones isolated from marine-derived *Aspergillus niger*

**Ghada Mahmoud Abdelwahab**[1,2]\*, **Amira Mira**[1], **Yuan-Bin Cheng**[3], **Tarek A. Abdelaziz**[4†], **Mohamed Farid I. Lahloub**[1], **Ashraf Taha Khalil**[1]

1 Department of Pharmacognosy, Mansoura University, Mansoura, Egypt, 2 Department of Pharmacognosy, Horus University, New Damietta, Egypt, 3 Department of Marine Biotechnology and Resources, National Sun Yat-sen University, Kaohsiung, Taiwan, 4 Marine Invertebrates, National Institute of Oceanography and Fisheries, Red Sea Branch, Hurghada, Egypt

† Deceased.
\* ghabdelwahab@horus.edu.eg

**Data Availability Statement:** All relevant data are within the manuscript and its Supporting information files.

## Abstract

*Aspergillus niger* metabolites exhibited a wide range of biological properties including antioxidant and neuro-protective effects and some physical properties as green synthesis of silver nanoparticles AgNP. The present study presents a novel evidence for the various biological activities of green synthesized AgNPs. For the first time, some isolated naphtho-γ-pyrones from marine-derived *Aspergillus niger*, flavasperone (**1**), rubrofusarin B (**2**), aurasperone A (**3**), fonsecinone A (**4**) in addition to one alkaloid aspernigrin A (**7**) were invistigated for their inhibitory activity of acetylcholine esterase AChE, a hallmark of Alzheimer's disease (AD). The ability to synthesize AgNPs by compounds **3**, **4** and **7** has been also tested for the first time. Green synthesized AgNPs were well-dispersed, and their size was ranging from 8–30 nm in diameter, their morphology was obviously spherical capped with the organic compounds. Further biological evaluation of their AChE inhibitory activity was compared to the parent compounds. AgNps dramatically increased the inhibitory activity of Compounds **4**, **3** and **7** by 84, 16 and 13 fold, respectively to be more potent than galanthamine as a positive control with $IC_{50}$ value of 1.43 compared to 0.089, 0.311 and 1.53 of AgNPs of Compounds **4**, **3** and **7**, respectively. Also compound **2** showed moderate inhibitory activity. This is could be probably explained by closer fitting to the active sites or the synergistic effect of the stabilized AgNPs by the organic compounds. These results, in addition to other intrinsic chemical and biological properties of naphtho-γ-pyrones, suggest that the latter could be further explored with a view towards other neuroprotective studies for alleviating AD.

**Funding:** The authors received no specific funding for this work.

**Competing interests:** The authors have declared that no competing interests exist.

## 1. Introduction

Alzheimer (AD) is a progressive multifactorial neurodegenerative disorder affecting millions of elders all over the world leading to increased burden on family members and medical care system. AD accounts for 80% of cases of dementia and is manifested by memory loss and cognitive impairment enough to interfere with daily life in addition to behavioral change [1]. Current treatment mitigates the symptoms rather than alter the course of disease [2, 3]. The disease has two hallmarks, first the deposition of β-amyloid peptide that clump into plaques that contribute The disease has two hallmarks, first the deposition of β-amyloid peptide that clump into plaques that contribute to aggregation of abnormal tau peptides forming tangles leading to disruption of transport system and damage of the neurons [4, 5], the insufficiency of acetylcholine neurotransmitter in some synapses of the brain cells leading to their degeneration [6, 7]. Consequently, the leading prescribed drugs are acetylcholine esterase (AChE) inhibitors as rivastigmine and donepezil may improve the symptoms. Recent research is ongoing (though with little success) on β-secretase inhibitors and vaccine that stimulate body's immune system to attack abnormal form of tau protein are still under investigation [2, 8, 9]. Reduction of oxidative stress and neuro-protection is another appealing target of therapy [10–12]. Besides its recent applications in industry [13], detection and diagnosis of some diseases [14], nanotechnology has a valuable role in drug delivery and monitoring [15, 16]. Multifunctional nanoparticles may help in early molecular diagnosis of AD through signal transduction approach by utilizing their special physical (optical, electric and magnetic), chemical and biological characteristics to transform a biological signal (a biomarker) to a recordable one [14], For delivery of drugs acting on central nervous system, various nanocarriers such as dendrimers, nano gels, liposomes, nano-emulsions, polymeric nano-particles and nano suspensions have been studied. The nanotreatment methods for AD include neuroprotection against toxicity of the oxidative stress of free radicals and amyloid-β-peptide (Aβ) oligomers as well as nanocarriers (particle size 1–100) for targeted drug delivery that can cross blood/brain barrier efficiently [17], The important agents for neuroprotection include nanogels, nano-cerium, nano-silver, dendrimers and gold nanoparticles [18]. The major nanocarriers include metal chelator nanocarriers (iron and copper chelators), cholinesterase inhibitors nanocarriers, acetylcholine nanocarrier, anti-oxidant nanocarriers, and gene nanocarriers [19, 20].

Among different nanometals, Silver nanoparticles (AgNPs) has acquired unlimited interest owing to their exceptional chemical stability, non-allergic properties, besides their significant antimicrobial, anti-inflammatory, and neuroprotective activities [21, 22]. Silver nanoparticles are used in wound dressings, disinfectant formulas, skin creams, contraceptive devices, surgical instruments and bone prostheses [23, 24]. Synthesis of nanoparticles frequently requires high temperature, pressure; utilizes toxic chemicals and is not cost-effective. The green method that uses micro-organisms or plants (or their secondary metabolites) is simple, quick, low-cost, ecofriendly and produce biocompatible, stable nano-particles [25]. Several plant extracts [22, 26], bacteria [27], cyanobacteria [28] and fungi [29, 30], notably *Aspergillus niger* produced AgNPs with size range down to 7.7 nanometersuggesting [31, 32]. Shape and size of nanoparticles may be adapted in order to deliver effective anti-AD drugs with reduced neurotoxicity [33–35].

The significant ability of *Aspergillus niger* to synthesize AgNPs may be due to its abundant bioactive reducing metabolites [36, 37]; some with neuroprotective activity [38]. *Aspergillus sp.* is generous producer of naphthopyrone phenolic compounds under certain stress conditions [39]. These compounds exhibited diverse biological properties as anti-bacterial [40, 41], hepatoprotective [42], Acyl CoA inhibitor [43], cytotoxic [44], anti-mutagenic [45], antiallergic [46], antiviral [47], non-toxic antioxidant [48], anti-xanthine oxidase [49], anti-tyrosinase

[47], anti-HIV-1 integrase [47], anti-COX-2 [50] and anti-AChE/anti-β-secretase [51] activities. Considering AD as a multifactorial disease, the inhibition of the latter enzyme may be potentiated by reducing oxidative stress secured by the inherent potent antioxidant activity, thus rendering naphtha-γ-pyrones worthy candidates for alleviating AD symptoms.

The current study describes green synthesis of AgNPs by pure isolated metabolites which resulted in synergy of its activity against AChE enzyme and may lead to possible decrease of the dose required to exert their action with lower side effects. This was done by using marine-derived *Aspergillus niger* solid fermentation culture extract to isolate and identify (by spectroscopic analysis) six naphthopyrones derivatives viz. flavasperone(**1**), rubrofusarin B (**2**), aurasperone A (**3**), fonsecinone A (**4**), asperpyrone B (**5**), asperpyrone C (**6**) in addition to one alkaloid aspernigrin A (**7**). To estimate the AChEI of the isolated compounds, Ellman's method was adopted [52]. The ability to green synthesize silver nanoparticles from aqueous solution of $AgNO_3$ was also investigated [53]. The produced capped silver nanoparticles were characterized using UV-Vis spectroscopy, transmission electron microscopy (TEM). Their AChEI activity was compared to their parent compounds using galanthamine as positive control.

## 2. Materials and methods

### 2.1. General experimental procedures

$^1$H and $^{13}$C NMR spectra were obtained in $CDCl_3$, $CDOD_3$ and DMSO-$d_6$ solutions with TMS as internal standard at 400 MHz for $^1$H NMR and 100 MHz for $^{13}$C NMR on *BRUKER Avance III* spectrometer. Mass experiments were conducted with LC/MS system (*Advion* compact mass spectrometer (CMS) NY,USA) by using Analyst version 1.4.1 software (*MDS Sciex*) where analytes were ionized in the negative and positive mode using electrospray ionization (ESI) interface. Final compounds purity was monitored by using pre-coated silica gel 60 GF$_{254}$ (20 x 20 cm, 0.2 mm thick) on aluminum sheets (*Merck*, Germany), and pre-coated reversed phase RP$_{18}$F$_{254S}$ (20 X 20 cm) plates, (*Merck*, Darmstadt, Germany), where as UV and vanillin/sulfuric acid spray reagent were used as revealing agents. Silica gel G 60–230 mesh (*Merck*, Darmstadt, Germany), Sephadex LH-20 (*Sigma-Aldrich*, Missouri, USA) were used for column chromatography. RP HPLC using Cosmosil AR-II, 250 x 10 mm i.d and 250x 4.6 mm i.d using a *Jasco PU2089* gradient pump and *PU2075 UV/VIS* detector. Reading absorbance was done by BioTek Microplate reader. HRTEM (JEM-2100) (JEOL, Tokyo, Japan) attached to a CCD camera at an accelerating voltage 200 kV was used for silver nanoparticles imaging and analytical characterization to assess their size, shape, and morphology.

### 2.2. Chemicals and reagents

Organic solvents were distilled before use and spectral grade solvents were used for spectroscopic measurements. All chemicals used were reagent grade and purchased from commercial suppliers. Reagents for biological assays were used as previously prescribed [54].

### 2.3. Fungal material, isolation and culture conditions

*Aspergillus niger* was isolated from the Red Sea tunicate *Phallusia nigra* collected by SCUBA diving at 5 m in depth (located at 27˚17'04.8"N, 33˚46'30.3"E) from a coral reef area in the Red Sea, Hurghada, Egypt in June 2017. The sample was collected and identified by researchers at the National Institute of Oceanography and Fisheries, Hurghada, Egypt, that are allowed to carry out their scientific research, including collecting samples from the field study area with no official permit (Act 4, 1994). Furthermore, this organism was recently listed among the

invasive species of the Mediterranean Sea [55]. Two organisms only were collected and their fresh weight was less than 100 gm. After rinsing the specimen with sterile sea water, it was transferred directly to the laboratory in a cooler bag filled with sterile sea water to be processed immediately.

The tunicate was rinsed three times with sterile seawater and superficially disinfected with 70% ethyl alcohol for 2 minutes then, cut aseptically into small pieces (2 x 2 cm) using a sterile dissection razor and cultivated on Potato Dextrose Agar (PDA) medium (*Oxoid* Ltd, Basingstoke, Hants, UK) plates in 50% aged sea water supplemented with 250 mg/L Amoxycillin to avoid any bacterial growth. The plates were incubated at room temperature for 1–2 weeks until adequate growth of the fungus. Pure strains of *Aspergillus niger* were isolated by repeated re-inoculation on saline PDA plates. Pure cultures were streaked on PDA slants for further study [56].

## 2.4. Identification of the endophytic isolate

The fungus was identified as *Aspergillus niger* by PCR using the universal fungal primers Internal Transcript Spacer regions (ITS1 and ITS4) (GenBank accession No.LC582533). A voucher strain of the fungus is deposited at the biological lab of the Pharmacognosy Department, Faculty of Pharmacy, Mansoura University, Egypt.

## 2.5. Cultivation

The fresh mycelia in each of thirty PDA petri dish were inoculated into an pre-autoclaved *Erlynmeyer* 1L conical flask containing 80 g wheat in 110 ml 50% aged sea water (30 flasks). The cotton-plugged fermentation flasks were incubated for three weeks at room temperature away from light.

## 2.6. Extraction and isolation of metabolites

Secondary metabolites of the fungus were extracted with EtOAc (3 x 1L) by sonication at 50˚C for 15 minutes. The combined EtOAc extract was filtered and concentrated to dryness using rotary vacuum evaporator at 50˚C followed by defatting by *n*-hexane in a separating funnel. The dried extract (6 g) was fractionated over CC (SiO$_2$, CH$_2$Cl$_2$/EtOAc 100: 0 to 0: 100, then EtOAc/MeOH 100: 0 to 0:100) to afford 15 fractions. Fr. 1–1 (46 mg) was eluted by methylene chloride (100%) to afford a mixture of compounds **1** and **2** that was rechromatographed over CC (SiO$_2$, petroleum ether/CH$_2$Cl$_2$, 20:80) to yield compound **1** (12 mg) and compound **2** (5 mg). Fr. 1–3 (26 mg) was eluted by CH$_2$Cl$_2$/EtOAc (90: 10) then, it was subjected to CC (SiO$_2$, CH$_2$Cl$_2$/EtOAc75: 25, isocratic) to afford compound **3** (17 mg) and compound **4** (7 mg), then each of compound **3, 4** were finally purified over Sephadex LH-20 using CH$_2$Cl$_2$/MeOH (85:15) as eluting solvent. Fr. 1–2 (128 mg) also containing compound **3** and **4** at nearly about 25% of their total weight by TLC visualization and still under investigation but, choosing fr. 1–3 was based on the purity of the fraction. Fr. 1–4 was eluted by CH$_2$Cl$_2$/EtOAc (85: 15) then, it was subjected to NP-HPLC Silica column, isocratic Hexane/EtOAc (1:2), 290 nm, flow 2.0 mL/min to afford three sub-fractions. Sub-fr. 1-4-1(23.8 mg) was further purified using RP-HPLC, *CN*-column using isocratic 60% MeOH, flow rate 2.0 mL/min, and detection at 210 nm, to yield two sub-fractions. Sub fraction 1-4-1-1 yielded compound **3** (14.8 mg) and Sub-fr.1-4-1-2 (3 mg) was finally purified over RP-HPLC, biphenyl column using isocratic 75% Acetonitile, flow rate 1.5 mL/min at 210 nm to afford compound **5** (0.5 mg). Sub-fr. 1-4-2 gave compound **4** (82.3 mg). Sub-fr. 1-4-3 yielded compound **6** (2.3 mg). Fr. 15 eluted by EtOAc 100% was purified by recrystalization to afford compound **7** (7.5 mg).

## 2.7. Synthesis of silver nanoparticles (AgNPs)

The isolated and identified compounds were screened for their ability towards green synthesis of AgNPs from silver nitrate solution. Therefore, five mM of each compound **3**, **4** and **7** were dissolved in 1 ml ethanol by heating and stirring. Six serial dilutions of each compound were prepared. Serial dilutions were used in order to detect the minimum concentration as well as the optimum concentration necessary for the synthesis of AgNPs. Two hundreds μl of each dilution were added to two adjacent wells in a 12-well plates (1st well for AgNPs and the 2nd well for blank), for each corresponding well 1 ml of $AgNO_3$ (2 mM) solution was added dropwise to the test wells, while 1ml of distilled water was added to the blank wells. Then, the microwell plates were heated to 50˚C in a water bath for 1 hour and the color changes were observed after cooling gradually from yellow to brown. The brown color indicates the formation of AgNPs by reducing $Ag^+$ to $Ag°$. The biosynthesized AgNPs were centrifuged, lyophilized and stored for further studies.

## 2.8. UV-visible spectroscopy

Reduction of silver ions in solution was monitored using the UV-Vis absorption as a method of choice. So herein Elisa plate reader was used as a high throughput screening method alongside with monitoring the color change. Absorption was read at 450 nm for each experiment after 60 min and after 48 h.

## 2.9. Transmission electron microscopy (TEM)

Size, shape, and morphology of the biosynthesized AgNPs were measured using a transmission electron microscope HRTEM (JEM-2100) (JEOL, Tokyo, Japan) at 200 kV. TEM grids were prepared by placing few drops of the AgNPs suspensions on carbon-coated copper grids and allowed to be slowly evaporated prior to recording the TEM images.

## 2.10. Molecular docking study

With the aim to investigate the different theoretical bindings of the protein-ligand geometries at molecular level, molecular docking experiments were performed using molecular operating environment (MOE) program version 2014 (0901). The aged phosphorylated AChE crystal structure was retrieved from RCSB-Protein Data Bank (PDB, code 1CFJ) with resolution 2.60 Å and imported into workspace of MOE. Protein energy was set up and hydrogens were added. The 3D structures of the ligands were drawn using Chem3D 15.0 software (Cambridge soft corporation, Cambridge, MA, USA) and saved as mol2 format. The ligands were docked at the largest cavity (size 174 A˚) detected by the program.

## 2.11. Acetylcholine esterase inhibitory assay

Acetylcholine esterase inhibitory activity of isolated compounds and their synthesized silver nanoparticles were assessed using Ellman's method as previously prescribed [54].

## 3. Results

All the compounds (Fig 1) were identified by spectroscopic methods ($^1$H, $^{13}$C NMR), mass spectral data and by matching with the previously published data [57–60].

*Flavasperone(1)*: Yellow needles; $^1$H-NMR (400 MHz, CDCl3): 2.41 (3H, s), 3.84 (3H, s), 3.88 (3H, s), 6.19 (1H, s), 6.31 (1H, d, *J* = 2.0 Hz), 6.49 (1H, d, *J* = 2.0 Hz), 6.77 (1H,s), 12.74 (1H,s). $^{13}$C-NMR (100 MHz, CDCl3): 182.8 (C-4), 166.6 (C-2), 161.4 (C-8), 159.0 (C-10), 156.6 (C-5), 155.8 (C-11), 141.2 (C-13), 110.2 (C-3), 108.8 (C-14), 105.7 (C-6), 104.8 (C-12),

**Fig 1. Structures of compounds isolated from *Aspergillus niger*.**

97.9(C-7), 96.9 (C-9), 55.8 (10-OCH3), 55.4 (8-OCH3), 20.5 (C-15). ESI-MS (negative-ion mode) $m/z$ 285.0 [M—H]$^-$; (calcd for $C_{16}H_{14}O_5$ 286.0841).

*Rubrofusarin B(2)*: Yellow needles; $^1$H-NMR (400 MHz, CDCl3): 2.3 (3H, s), 3.85 (3H, s), 3.93 (3H, s), 5.93 (1H, s), 6.33 (1H, d, $J$ = 2.0 Hz), 6.52 (1H, d, $J$ = 2.0 Hz), 6.9 (1H, s), 14.91 (1H, s). ESI-MS (negative-ion mode) $m/z$ 285.0 [M—H]$^-$; (calcd for $C_{16}H_{14}O_5$ 286.0841).

*Aurasperone A(3)*: Yellow powder; $^1$H-NMR (400 MHz, CDCl3): 2.05 (3H, s), 2.35 (3H, s), 3.39 (3H, s), 3.55 (3H, s), 3.72 (3H, s), 3.96 (3H, s), 5.92 (1H, s), 5.99 (1H, s), 6.14 (1H, d, $J$ = 2.0 Hz), 6.35 (1H, d, $J$ = 2.0 Hz), 6.97 (1H, s), 7.09 (1H, s), 14.77 (1H, s), 15.19 (1H, s).$^{13}$C-NMR (100 MHz, CDCl3): 184.6 (C-4′), 184.5 (C-4), 167.7 (C-2), 167.6 (C-2′), 162.7 (5′-OH), 162.0 (5-OH), 161.4 (C-8′), 161.0 (C-6′), 160.2 (C-8), 158.5 (C-6), 153.4 (C-10a), 150.8 (C-10′a), 140.7 (C-9a), 140.6 (C-9′a), 117.6 (C-7), 111.4 (C-5a), 108.6 (C-5′a), 107.5 (C-3), 107.3 (C-3′), 105.2 (C-10′), 104.7 (C-4a), 104.3 (C-4′a), 101.4 (C-9), 101.2 (C-10), 96.9 (C-7′), 96.5 (C-9′), 62.1 (6-OCH$_3$), 56.2 (6′-OCH$_3$), 56.0 (8-OCH$_3$), 55.2 (8′-OCH$_3$), 20.8 (2-CH$_3$), 20.6 (2′-CH$_3$). ESI-MS (positive-ion mode) $m/z$ 571.16 [M + H]$^+$; (calcd for $C_{32}H_{26}O_{10}$, 570.1560).

*Fonsecinone A(4)*: Yellow powder; $^1$H-NMR (400 MHz, CDCl3): 2.06 (3H, s), 2.42 (3H, s), 3.36 (3H, s), 3.55 (3H, s), 3.72 (3H, s), 3.97 (3H, s), 5.94 (1H, s), 6.12 (1H, d, $J$ = 2.0 Hz), 6.27 (1H, s), 6.36 (1H, d, $J$ = 2.0 Hz), 6.90 (1H, s), 6.99 (1H, s), 12.78 (1H, s), 15.20 (1H,

s).[13]C-NMR (100 MHz, CDCl3): 184.6 (C-4'), 182.9 (C-4), 167.5 (C-2), 166.9 (C-2'), 162.8 (C-5'), 161.6 (C-6'), 161.1 (C-6'), 160 (C-8), 156.9 (C-10), 156.7 (C-5), 155.1 (C-10b), 150.8 (C-10'a), 140.8 (C-6a), 140.6 (C-9'a), 117.1(C-9), 110.7 (C-3), 109.4 (C-4a), 108.6 (C-5'a), 108.0 (C-10a), 107.4 (C-3'), 106.0 (C-6), 105.1 (C-10'), 104.2 (C-4'a), 101.6 (C-7), 97.0 (C-7'), 96.3 (C-9'), 61.2 (C-10-OCH$_3$), 56.2 (C-6'-OCH$_3$), 56,0 (C-8-OCH$_3$), 55.2 (C-8'-OCH$_3$), 20.7 (2'-CH$_3$), 20.6 (2-CH$_3$). ESI-MS (positive-ion mode) *m/z* 571.16 [M + H]$^+$; (calcd for C$_{31}$H$_{26}$O$_{10}$, 570.1560).

*Asperpyrone B*(**5**): Yellow powder; [1]H-NMR (400 MHz, CDCl3):6.29 (1H, s), 7.02 (1H, s), 6.96 (1H, s), 2.46 (3H, s), 12.79 (1H, s), 3.78 (3H, s), 3.59 (3H, s), 6.31 (1H, s), 6.18 (1H, d, *J* = 2.1), 6.42 (1H, d, *J* = 2.1), 2.53 (3H, s), 13.18 (1H, s), 3.59 (3H, s), 3.99 (3H, s).[13]C-NMR (100 MHz, CDCl3):183.0 (C-4'), 182.9 (C-4), 166.8 (C-2), 166.5 (C-2'), 161.6 (C-8'), 160.0 (C-8), 159.5 (C-10'), 156.6 (C-10), 156.5 (C-5), 155.8 (C-10'b), 155.0 (C-10b), 154.3 (C-5'), 140.9 (C-6a'), 140.7 (C-6a), 117.9 (C-9), 110.6 (C-3), 110.3 (C-3'), 109.7 (C-6'), 109.3 (C-4a), 108.5 (C-4'a), 108.1 (C-10a), 106.2 (C-6), 105.1 (10'a), 101.9 (C-7), 96.8 (C-9'), 96.4 (C-7'), 61.5 (C-8'-OCH$_3$), 56.1 (C-8-OCH$_3$), 56.0 (C-10'-OCH$_3$), 55.2 (C-10-OCH$_3$), 20.6 (2'-CH$_3$), 20.5 (2-CH$_3$). ESI-MS (positive-ion mode) *m/z*571.08 [M + H]$^+$; (calcd for C$_{31}$H$_{26}$O$_{10}$, 570. 1560).

*Asperpyrone C* (**6**): Yellow powder; [1]H-NMR (400 MHz, CDCl3): 2.12 (3H, s), 2.42 (3H, s), 3.46 (3H, s), 3.62 (3H, s), 3.79 (3H, s), 4.03 (3H, s), 5.98 (1H, s), 6.06 (1 H, d, *J* = 2.2 Hz), 6.21 (1H, s), 6.41 (1H, d, *J* = 2.2 Hz), 6.97 (1H, s), 7.15 (1H, s). ESI-MS (positive-ion mode) *m/z*571.15 [M + H]$^+$; (calcd for C$_{32}$H$_{26}$O$_{10}$, 570.1560).

*Aspernigrin A* (**7**):Colorless needles; [1]H-NMR (400 MHz, CDCl3): 3.9 (2H, s), 6.23 (1H, s), 7.27 (1H, m), 7.28 (2H, m), 7.34 (2H, m), 8.33 (1H, s), 14-NH$_2$ 9.52 (br s), 7.41(br s). [13]C-NMR (100 MHz, CDCl3): 187.1 (C-4), 166 (14-NH$_2$), 151.3 (C-6), 142.2 (C-1), 137.4 (C-8), 129.5 (C-10), 129.3 (C-12), 129.2 (C-9), 129 (C-13), 127.4 (C-11), 118.8 (C-5), 117.9 (C-3), 38.2 (C-7). ESI-MS (negative-ion mode) *m/z* 227 [M—H]$^-$; (calcd for C$_{13}$H$_{12}$N$_2$O$_2$Na 251.0796).

## 3.1. Synthesis of silver nanoparticles (AgNPs) and characterization by UV-vis spectroscopy

Aurasperone A, fonsecinone A and aspernigrin A were screened as reported by Chauhan, 2012 [53] with some modifications to get a rapid tool for screening many concentrations of different compounds for their ability to green synthesize AgNPs. Reading absorbance spectro-photometrically using 12-microwell plates at 450 nm is an easy method as, transparency of the microwell plate facilitate the color monitoring. Asperpyrone B and asperpyrone C have small amounts that were enough only for spectroscopic identification while flavasperoneand rubro-fusarin B have a low polarity hence, these compounds could not be tested. (Fig 2) showed the absorption peak of the AgNPs synthesized by the three organic compounds obsedved between 441 and 463 nm as a result of their Surface Plasmon Resonance (SPR). Five mM of each tested compound was dissolved in 1 ml EtOH by heating and stirring. Six serial dilutions of each compound were prepared then 1 ml of 2 mM (AgNO$_3$) solution was added dropwise. Micro-well plates were heated to 50˚C in a water bath for 1 hour until color change to dark yellow. Figs 3 and 4 show the color change of the reaction solution from colorless (aspernigrin A) and pale yellow (aurasperone A, fonsecinone A) to faint brown after 1 h when compared with blank for each corresponding compound. The color was changed to dark brown when full reduction of silver ions was completed after 48 h at 25˚C. It was noted that even very small concentrations could synthesize AgNPs. The biosynthesized AgNPs were centrifuged lyophi-lized and tested for AChEI activity.

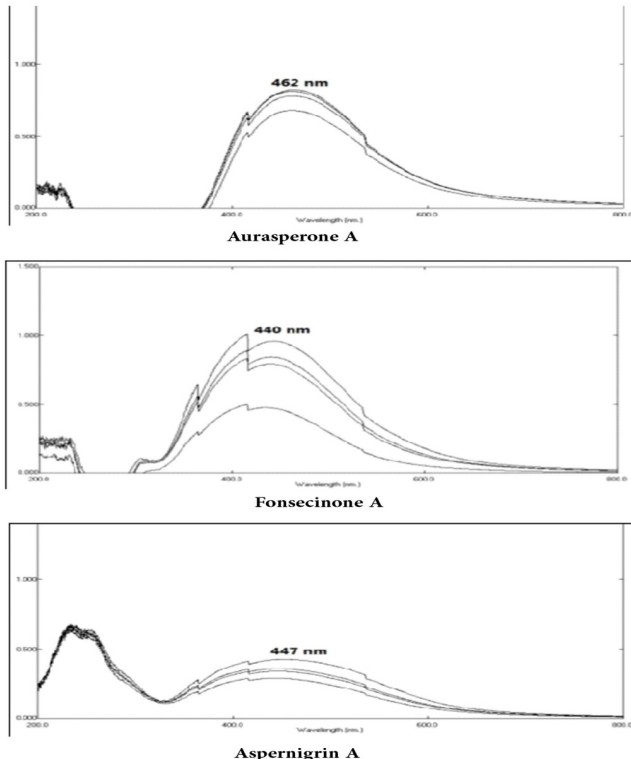

**Fig 2. UV–Vis spectrum of AgNPs synthesized using aurasperone A, fonsecinone A and aspernigrin A.**

## 3.2. Transmission electron microscopy (TEM) imaging

Transmission Electron Microscopy images of the biosynthesized AgNPs by aurasperone A, fonsecinone A and aspernigrin Awith optimum concentration in different magnifications are shown in (Fig 5). The AgNPs were well-dispersed, and their size was ranging from 8–30 nm in

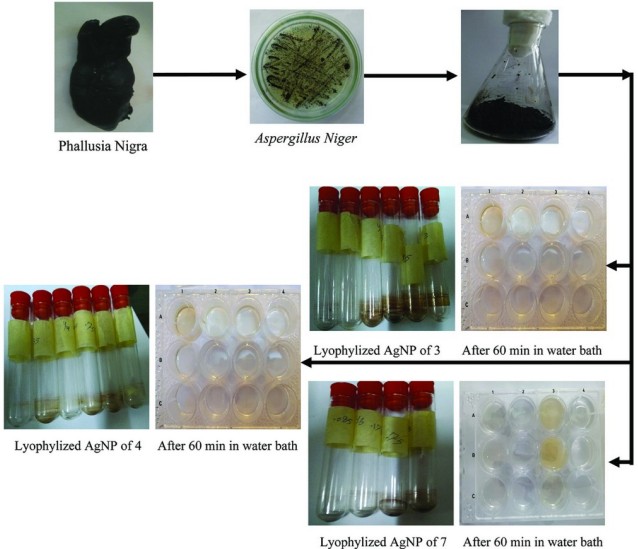

**Fig 3. Schematic representation of green synthesis of AgNPs by pure isolated compounds.**

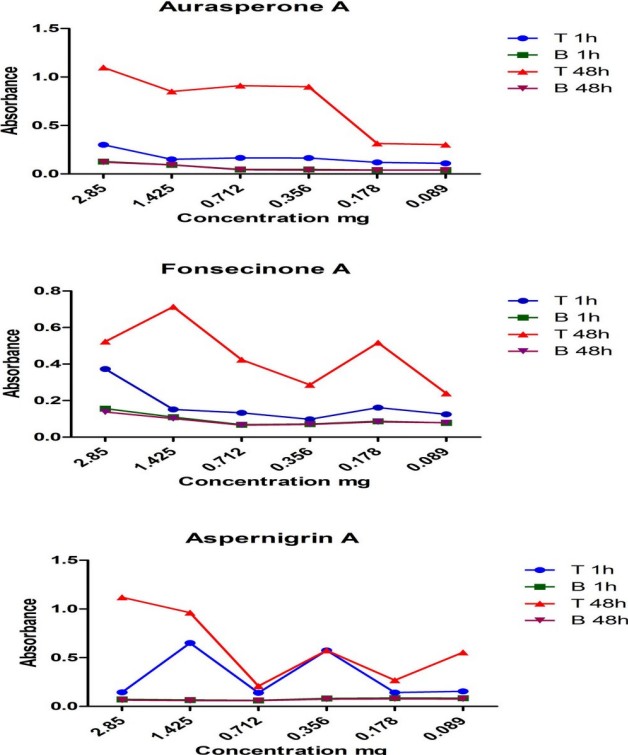

**Fig 4. UV absorbance of compounds aurasperone A, fonsecinone A and aspernigrin A upon addition of AgNO₃ solution after 1 h. and after 48 h.**

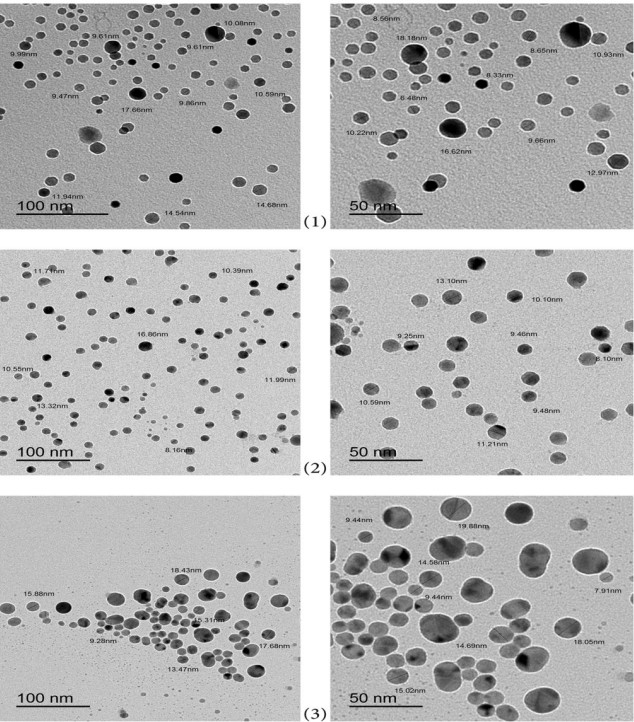

**Fig 5. Transmission electron microscopy images of biosynthesized AgNPs at different magnifications of (1) Aurasperone A; (2) Fonsecinone A; (3) Aspernigrin A.**

**Table 1. AChEI activity of the isolated compounds and their synthesized AgNPs[1].**

| Compound | $IC_{50}(\mu M)$ | Binding energy (Kcal/mol) | H-Bonding | Arene-H Bonding |
|---|---|---|---|---|
| AgNPs | 2.29 ± 0.013 | - - - - - - - | - - - - - - - | - - - - - - - |
| Flavasperone (1) | > 25 inactive | -6.02967262 | Tyr-130,Trp -84 | - - - - - - - |
| Rubrofusarin B (2) | 13.87 ± 2.16 | -11.7267284 | Glu-199, Ser-122 | - - - - - - - |
| Aurasperone A (3) | 4.90 ± 2.16 | -14.2922001 | Asn-85,Asp-72,Phe-331 | Phe-331 |
| Aurasperone A AgNPs | 0.311 ± 0.018 | - - - - - - - | - - - - - - - | - - - - - - - |
| Fonsecinone A (4) | 7.52 ± 2.16 | -11.1807451 | Tyr-70,Phe-288, Lie-287 | Tyr-334 |
| Fonsecinone A AgNPs | 0.089 ± 0.005 | - - - - - - - | - - - - - - - | - - - - - - - |
| Asperginin A (7) | 20.17 ± 3.03 | -10.181241 | Phe-288, Phe-330 | Phe-330 |
| Asperginin A AgNPs | 1.53 ± 0.076 | - - - - - - - | - - - - - - - | - - - - - - - |

[1] Galanthamine (positive control) $IC_{50}$ = 1.43 ± 0.36 μM, Binding score -11.8111763, H-Bonding (Gly-118, Gly-119, His-440) and Hydrophobic H-Bonding (Phe-330).

diameter. In addition, the morphology of the synthesized AgNPs was obviously spherical capped with the organic compounds which indicates a good stabilization effect of the investigated compounds [61].

### 3.3. Acetylcholine esterase inhibitory assay

Acetylcholine esterase inhibitory (AChEI) activity was assessed by using Ellman's method [52, 62, 63]. The results are shown in Table 1. Asperpyrone-type bis-naphtho-γ-pyrones showed remarkable AChE inhibition particularly with their green synthesized AgNPs. The AgNPs showed AChE inhibitory activity with $IC_{50}$ value of 2.29 μM and has increased the AChE inhibitory activity of fonsecinone A by 84 fold ($IC_{50}$ value decreased from 7.52 to 0.089μM) followed by aurasperone A AgNPs which its inhibitory activity has been increased by 16 fold ($IC_{50}$ value decreased from 4.9 to 0.311 μM) compared to galanthamine as a positive control ($IC_{50}$ values of 1.43). The activity of the alkaloid aspernigrin A also increased by 13 fold after AgNPs synthesis ($IC_{50}$ value decreased from 20.17 to 1.53 μM). This remarkable decrease in the $IC_{50}$ value could be ascribed to the synergistic effect of AgNPs when capped with the tested compounds. The linear naphtho-γ-pyronerubrofusarin B showed moderate inhibitory activity with $IC_{50}$ value of 13.87 μM, while compound flavasperone didn't show any inhibitory activity.

## 4. Discussion

### 4.1. Synthesis of silver nanoparticles (AgNPs)

AgNPs are known to exhibit a UV–Visible absorption maxima in the range of 400–500 nm due to their Surface Plasmon Resonance (SPR) [64, 65]. SPR is the collective oscillation of conduction band electrons which are in resonance with the oscillating electric field of incident light, that produce energetic plasmonic electrons through non-radiative excitation [66]. It is the basis of many standard tools for measuring adsorption of material onto planar metal or metal nanoparticle surfaces. Aurasperone A, fonsecinone A and aspernigrin A showed absorption peaksappearing between 441 and 463 nm because of their SPR may differ due to variation in the size and shape of the synthesized AgNPs. The synthesis of AgNPs was detected by the color change to dark yellow or brown color, which is a typical color of AgNPs in solution [53].

### 4.2. Acetylcholine esterase inhibitory assay

To gain insights about the theoretical binding modes of the active compound with the binding residues of the active sites of AChE, molecular docking simulation experiments were done and

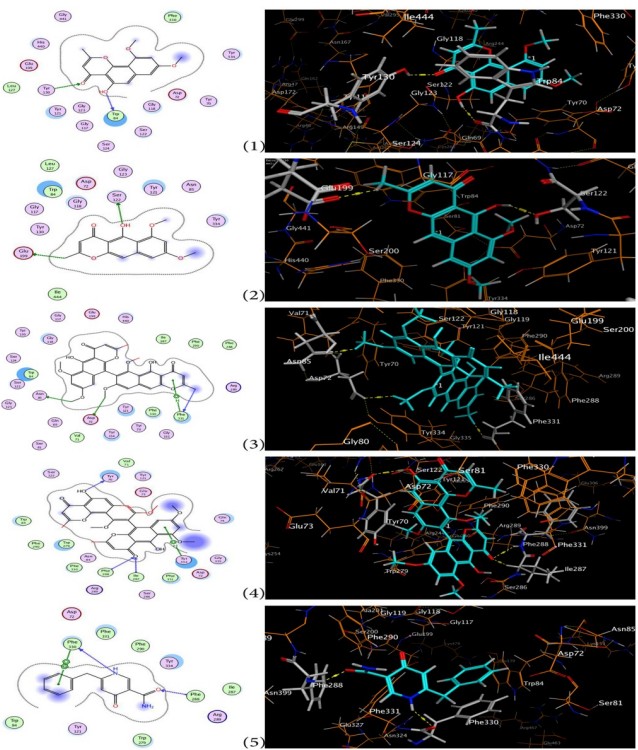

**Fig 6. Binding residues of (1) flavasperone, (2) rubrofusarin B, (3) aurasperone A, (4) fonsecinone A and (5) aspernigrin A in AChE assay using MOE program.** Dashed yellow lines indicate H-bonds in 3D ligand interaction diagram and blue ones in 2D diagram. Carbons are in turquoise, nitrogens in blue, and oxygens in red.

the results were compatible with AChE inhibitory assay results (binding energy is shown in Table 1). Kinetic studies, torpedo and mammal x-ray crystallography of AChE were showed that the active sites consist of the esteratic site, the active narrow gorge, the peripheral anionic site (PAS), the oxyanionic hole and the catalytic anionic site. The esteratic site which contains 5 residues: the catalytic triad (Ser-200, Glu-327, His-440), Phe-288 and Phe-290. This esteratic site lies at the bottom of the active narrow gorge which is nearly about 20 A° long and 14 aromatic residues line a substantial portion (40%) of the gorge surface among them (Trp-84, Tyr-130, Phe-330, Phe-331). PAS is containing 5 residues (Tyr-70, Asp-72, Tyr-121, Trp-279, Tyr-334). Binding to PAS causes conformational change of AChE and prevents the passage of AChE through the narrow gorge. Another important site is the oxyanion hole which has (Gly-118, Gly-119 and Ala 201) residues that plays a remarkable role in high-energy intermediates stabilization and the transition state by hydrogen bond [63, 67]. Furthermore the catalytic anionic site which has (Asn-85, Ser-122, Glu-199) residues [68]. As shown in (Fig 6), aurasperone A has the highest docking score (-14.29 Kcal/mol) and could bind through three hydrogen bonds between methoxy group at C-8 with Asp-72, methoxy group at C-8' with Asn-85 and methyl group at C-2 with Phe-331, in addition to hydrophobic-H bond with Phe-331. Fonsecinone A which is an isomer of aurasperone A has a lower docking score (-11.18 Kcal/mol) which may be due to difference in the arrangement of pyrone at naphthalene ring. Fonsecinone A could have three hydrogen bonds with the active sites one between hydroxyl group at C-5 with Tyr-70. The second is between carbonyl group at C-4' with Phe-288 and Lie-287, in addition to hydrophobic-H bond with Tyr-334. Rubrofusarin B has a docking score (-11.73 Kcal/mol) could bind by only two hydrogen bonds between hydroxyl group at C-5 with

Ser-122 and methyl group at C-2 with Glu-199. The absence of binding to PAS, active site gorge of rubrofusarin B and formation of only two hydrogen bonds may explain the decrease in the docking score of rubrofusarin B. Aspernigrin A has the least docking score (10.18 Kcal/mol). It could bind by two hydrogen bonds between the carbonyl group at C-14 with Phe-288 and the *N*-atom with Phe-330, also hydrophobic-H bond with Phe-330 has observed. Lacking of naphtho-γ-pyrone and two hydrogen bonds only may account for the low activity of aspernigrin A.

Some features of structure/activity relationship (SAR) of the isolated compounds could be included as follow; bis-naphtho-γ-pyrones are more active than naphtho-γ-pyrones [37, 69], linear naphtho-γ-pyrones are more active than angular naphtho-γ-pyrones, presence of methoxy group at C-8, 8', hydroxyl group at C-5 and methyl group at C-2 seem essential for AChE inhibition. Our results were consistent with previous studies regarding the correlation between methoxy group at C-6, C-7, C-8 and the arrangement of pyrone at naphthalene ring [51]. It is worthy to note that carbonyl with neighboring phenolic OH in these compounds can provide them with antioxidant as well as ability to chelate some heavy metals as predicted from behavior of other natural compounds sharing similar pyrone ring as flavones and xanthones. This probably could support their potential as promising anti-AD agents [70, 71].

The mechanism by which AgNPs inhibit AChE is believed to be due to structural perturbation of the enzyme. This inhibition may be ascribed to AChE adsorption on nanoparticle surface and subsequent bracing of enzyme structure. Change in the distribution of surface charge on the enzyme and induction of $H_2S$ synthesizing enzymes can also contribute to the AChE inhibition [72, 73]. Most proteins have strong adsorption at the solid–water interface, and adsorption of enzymes on AgNPs can result in inactivation due to conformational change [74]. Despite some reports of *in vivo* neurotoxicity of AgNPs probably through alteration of the permeability of blood brain barrier cells thereby stimulating the oxidative stress in the nerve cells [75–77]; their mechanism is still not fully understood. Several metallic nanoparticles were reported to exhibit *in vivo* neuroprotective and memory enhancing activity without noticeable toxicity such as gold [78], iron [79], silver [22]. Gold nanoparticles, in particular, were proved to reduce accumulation of amyloid β-peptides, while iron nanoparticles reduced tau protein aggregation. Optimizing AgNPs by reducing the size to 0.1 nm (1 A˚) revealed better biological activity and lower toxicity [80]. The combined action of AgNps with naphthopyrones that have favourable AChEI, antioxidant anti-inflammatory, antimutagenic and chelating activities could open up a new horizon for possible development of drugs for alleviation of Alzheimer's disease.

## 5. Conclusion

The present study displayed the ability of aurasperone A, fonsecinone A and aspernigrin Ato synthesize spherical, stable, and well-dispersed AgNPs with size ranging from 8–30 nm in diameter. They were coated with the isolated natural compounds. AgNPs dramatically enhanced the inhibitory activity of aurasperone A, fonsecinone A and aspernigrin Aon AChE more efficiently than the other examined derivatives. Further investigation of the *in vitro* neuroprotective properties of the naphtho-γ-pyrones dimers is suggested to augment the profile of their potential anti-AD activity. It is meaningful to harness the fungal ability to produce sustainable, abundant naphtho-γ-pyrone compounds and further study their *in vivo* activities before and after coating/combining with proper nano-metallic particles. These natural compounds with renewable fungal source possess several beneficial structural and biological characteristics, thus rendering them as promising multi-target potential drugs for Alzheimer's disease.

## Supporting information

**S1 File.**
(PDF)

## Acknowledgments

We would like to thank Dr. Mohamed El-Metwally for his help in collecting and identifying the marine specimens. [Prof. Dr/ Tarek A. Abdelaziz] passed away before the submission of the final version of this manuscript. [Ghada Mahmoud Abdelwahab] accepts responsibility for the integrity and validity of the data collected and analyzed.

## Author Contributions

**Data curation:** Ghada Mahmoud Abdelwahab, Ashraf Taha Khalil.

**Formal analysis:** Ghada Mahmoud Abdelwahab.

**Methodology:** Ghada Mahmoud Abdelwahab, Amira Mira, Yuan-Bin Cheng, Ashraf Taha Khalil.

**Resources:** Tarek A. Abdelaziz.

**Software:** Ghada Mahmoud Abdelwahab.

**Supervision:** Mohamed Farid I. Lahloub, Ashraf Taha Khalil.

**Visualization:** Ghada Mahmoud Abdelwahab.

**Writing – original draft:** Ghada Mahmoud Abdelwahab.

**Writing – review & editing:** Amira Mira, Mohamed Farid I. Lahloub, Ashraf Taha Khalil.

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
