## [Decision Letter · Decision Letter 0]

1 Apr 2021

PONE-D-21-05273

Acetylcholine esterase inhibitory activity of green synthesized nanosilver by naphthopyrones isolated from marine-derived Aspergillus Niger

PLOS ONE

Dear Dr. Abdelwahab,

Thank you for submitting your manuscript to PLOS ONE. After careful consideration, we feel that it has merit but does not fully meet PLOS ONE’s publication criteria as it currently stands. Therefore, we invite you to submit a revised version of the manuscript that CAREFULLY addresses the points raised during the review process. Referees might have suggested some literature citations, authors are free to decide upon.

We look forward to receiving your revised manuscript.

Kind regards,

Yogendra Kumar Mishra, Ph. D.

Academic Editor

PLOS ONE

Journal Requirements:

3. In your Methods section, please provide additional location information, including geographic coordinates for the data set if available.

Reviewers' comments:

Reviewer's Responses to Questions

**Comments to the Author**

1. Is the manuscript technically sound, and do the data support the conclusions?

Reviewer #1: Yes

Reviewer #2: Partly

2. Has the statistical analysis been performed appropriately and rigorously? 

Reviewer #1: Yes

Reviewer #2: No

3. Have the authors made all data underlying the findings in their manuscript fully available?

Reviewer #1: Yes

Reviewer #2: Yes

4. Is the manuscript presented in an intelligible fashion and written in standard English?

Reviewer #1: Yes

Reviewer #2: No

5. Review Comments to the Author

Reviewer #1: In the present work Abdelwahab et. Al. described the separation and characterization of six naphtho-�-pyrones (2 monomers and 4 dimers) and one pyridine-4-one alkaloid from solid culture of marine-derived Aspergillus niger for application in acetylcholine esterase inhibitory activity (AChEI). Further Ag nanoparticles were also synthesized and tested for AChEI. The results are interesting. However, there are some defects needed to be further revised. Including:

1. Characterization of Ag NPs are poor. Authors should provide XRD, Raman and XPS analysis for proper characterization.

2. There is absent of mechanism interpretation and analysis to support the conclusion

3. The quality of ALL OF FIGURES should be further decorated. The following literature should be consulted and employed:

4. The English should be carefully polished. Also, many format problems should be addressed.

ACS Biomaterials Science & Engineering, 2020, 6, 5527–5537; Theranostics, 2020, 10, 7841-7856; Journal of Biomedical Nanotechnology, 2020, 16, 283-303.

Reviewer #2: The manuscript entitiled “Acetylcholine esterase inhibitory activity of green synthesized nanosilver by naphthopyrones isolated from marine-derived Aspergillus Niger” submitted by Abdelwahab et al is an nice submission regarding advancement in nanotechnology. The idea , concept and the methodology of the manuscript is novel as well as well defined however as manuscript , it consist of many flaws that needs to be corrected before acceptance. The comments are :

1. The whole manuscript should be screened and rewritten with proper English. The quality of the English is really poor which is ruining the quality of the work.

2. Abstract needs to be rewritten defining the novelty of work with some quantitative data.

3. The first para of the introduction is full of typos and grammatical error. Moreover, the content are rewritten. `This paragraph must be rewritten with a full formatting.

4. Paragraph 3, Write the hypothesis of the work and the novelty.

5. Number of recent manuscript has been published in silver nanoparticles, Authors should cite the recent papers. Some of the suggestion are in the last comment.

6. Don’t keep the section 2.3. It looks odd to give heading and saying it has been mentioned previously.

7. Combine section 2.4 and 2.5 under one subheading.

8. Section 2.9, use proper annotations and numerical.

9. Materials and methods section needs to be written in more details.

10. Please don’t use compound number in result section. Use the full name or an short name of the compound.

11. Table 1 needs to be presented as an graph. It should be not presented in this way. What was the SPR peak of the scan?

12. Figure 2 needs to be redrawn. The resolution is too poor.

13. Table 2 is not OK. Description and data is not matching.

14. Figure 4 is not okay.. resolution needs to be improved.

15. Where is the discussion? The authors has no where defined and discussed the significance of their finding. It seems that the manuscript is incomplete.

16. Lastly, the authors are suggested to cite recent papers like DOI: 10.1039/C7RA05943D, https://doi.org/10.1093/toxsci/kfx204, https://doi.org/10.1016/j.msec.2018.07.037,https://doi.org/10.1080/21691401.2018.1503598 , DOI: 10.1016/j.msec.2021.111888 , DOI: 10.1016/j.bioorg.2020.104535 , DOI: 10.2217/nnm-2020-0138, DOI: 10.1016/j.mtchem.2020.100299; DOI: 10.1016/j.mtchem.2020.100345; https://doi.org/10.1021/acsomega.7b01522

6. PLOS authors have the option to publish the peer review history of their article (what does this mean?). If published, this will include your full peer review and any attached files.

Reviewer #1: No

Reviewer #2: No

---

## [Author Response · Author response to Decision Letter 0]

18 Jun 2021

Reviewer #1:.

1- (a) X-ray diffraction (XRD) analysis is used to check the crystalline nature of silver nanoparticles. So, it should be done on the dried powder. In our case, after lyophilization, it was a sticky material at the end of the wasserman test tube (as seen in Figure 3) because the amount of starting compound entering the synthesis was very little so, it cannot be in a crystal or powder form. In addition, other published papers performed (XRD) analysis because they used total extract in green synthesis of silver nanoparticles. Consequently, they have large amount but in our experiment we used pure isolated compounds with little yield.

(b) XPS is used for elemental analysis using solid surface and Raman is used for chemical structure identification as well as detection of contamination and impurities and give information about phase and polymorphism. In case of silver nanoparticles synthesized by total extract, Raman and XPS were performed as the total extract may contain different organic metabolites and other elements as impurities but here we used pure organic compounds which were isolated by pure organic solvents as proved by their NMR data attached in the supplementary file. Therefore, the probability of the presence of other elements is almost non-existent. Also these analyses are too expensive in our country due to low facilities and lack of funding. There is only one available equipment and it is reserved for several months.

2- Regarding the mechanism interpretation and analysis to support the conclusion; the effect of silver nanoparticles alone on acetylcholine esterase enzyme was performed and its IC50 is 2.29 ± 0.013 uM/ml. This confirms that the pure isolated organic compounds have synergistic effect on acetylcholine esterase inhibition after capping with silver nanoparticles possibly by stabilizing them for a longer time to exert its effect.

3- The quality of all figures was improved and resolution was adjusted between 300-600 dpi to meet PLOS ONE's style figure requirements.

Reviewer #2:

1- The whole manuscript was screened and some were rewritten with proper English.

2- Abstract was rewritten defining the novelty of work with some quantitative data.

3- The first paragraph was revised with a full formatting.

4- In paragraph 3, the hypothesis and the novelty of the work were added.

5- Some recent papers of your suggestions were added.

6- Section 2.3 was added to section 2.2.

7- Section 2.5 was combined with section 2.4 under one sub-heading.

8- Proper annotations were used as possible as we can.

9- Some details were added in Methods section.

10- The compound number was replaced by the full name of the compound in the result section.

11- Table 1 presented as a graph in Figure 4. The SPR was between 440 to 462 nm and was added to the manuscript.

12- Figure 2 was changed to Figure 3 and was re-drawn. The resolution was adjusted between 300-600 dpi to meet PLOS ONE's style figure requirements.

13- Table 2 became Table 1 and was revised.

14- Figure 4 became Figure 6 and was re-drawn. The resolution was adjusted between 300-600 dpi to meet PLOS ONE's style figure requirements.

15- Discussion was revised, clarified and significance of the findings was discussed in section 3.3

16- Recent papers were cited.

---

## [Decision Letter · Decision Letter 1]

16 Jul 2021

PONE-D-21-05273R1

Acetylcholine esterase inhibitory activity of green synthesized nanosilver by naphthopyrones isolated from marine-derived Aspergillus niger

PLOS ONE

Dear Dr. Abdelwahab,

Thank you for submitting your manuscript to PLOS ONE. After careful consideration, we feel that it has merit but does not fully meet PLOS ONE’s publication criteria as it currently stands. Therefore, we invite you to submit a revised version of the manuscript that addresses the points raised during the review process.

We look forward to receiving your revised manuscript.

Kind regards,

Mohammad Shahid, Ph.D.

Academic Editor

PLOS ONE

Reviewers' comments:

Reviewer's Responses to Questions

**Comments to the Author**

1. If the authors have adequately addressed your comments raised in a previous round of review and you feel that this manuscript is now acceptable for publication, you may indicate that here to bypass the “Comments to the Author” section, enter your conflict of interest statement in the “Confidential to Editor” section, and submit your "Accept" recommendation.

Reviewer #2: (No Response)

2. Is the manuscript technically sound, and do the data support the conclusions?

Reviewer #2: Partly

3. Has the statistical analysis been performed appropriately and rigorously? 

Reviewer #2: No

4. Have the authors made all data underlying the findings in their manuscript fully available?

Reviewer #2: Yes

5. Is the manuscript presented in an intelligible fashion and written in standard English?

Reviewer #2: Yes

6. Review Comments to the Author

Reviewer #2: The author has tried to response the queries well however, the discussion part is still not convincing. Authors are suggested to describe the results in a how and why pattern. Right now , only results are there. I will suggest them to separate the result and discussion section and provide a detail discussion.

In my opinion, its a crucial part for any manuscript and can't be ignored. Manuscript may be accepted after that followed by review.

7. PLOS authors have the option to publish the peer review history of their article (what does this mean?). If published, this will include your full peer review and any attached files.

Reviewer #2: No

---

## [Author Response · Author response to Decision Letter 1]

21 Aug 2021

Response to reviewers

1- Adressed to reviewer #2. (Previous round).

2- Adressed to reviewer #2. (Technically sound).

3- Adressed to reviewer #2. (Statistical analysis).

4- Adressed to reviewer #2. (Data availability).

5- Adressed to reviewer #2. (Standard English).

6- The reviewer suggested a more focus on discussion part and separate it from results which were actullay did. The results (P.6) and discussion (P.9) section was separated and the results were described in how and why pattern. Really, it is a crucial part. Thank you for your opinion and effort.

7- This item also was addressed to reviewer #2.

We did our maximum efforts to improve the manuscript guided by editor and reviewers' comments and we hope that the final version will be satisfactory for respected editor and reviewers.

Thank you

---

## [Decision Letter · Decision Letter 2]

24 Aug 2021

Acetylcholine esterase inhibitory activity of green synthesized nanosilver by naphthopyrones isolated from marine-derived Aspergillus niger

PONE-D-21-05273R2

Dear Dr. Abdelwahab,

We’re pleased to inform you that your manuscript has been judged scientifically suitable for publication and will be formally accepted for publication once it meets all outstanding technical requirements.

Kind regards,

Mohammad Shahid, Ph.D.

Academic Editor

PLOS ONE

Reviewers' comments:

Reviewer's Responses to Questions

**Comments to the Author**

1. If the authors have adequately addressed your comments raised in a previous round of review and you feel that this manuscript is now acceptable for publication, you may indicate that here to bypass the “Comments to the Author” section, enter your conflict of interest statement in the “Confidential to Editor” section, and submit your "Accept" recommendation.

Reviewer #2: All comments have been addressed

2. Is the manuscript technically sound, and do the data support the conclusions?

Reviewer #2: Yes

3. Has the statistical analysis been performed appropriately and rigorously? 

Reviewer #2: Yes

4. Have the authors made all data underlying the findings in their manuscript fully available?

Reviewer #2: Yes

5. Is the manuscript presented in an intelligible fashion and written in standard English?

Reviewer #2: Yes

6. Review Comments to the Author

Reviewer #2: The authors have addressed now all the queries and nicely revised the manuscript. The manuscript can now be accepted followed by the supervision of editor.

7. PLOS authors have the option to publish the peer review history of their article (what does this mean?). If published, this will include your full peer review and any attached files.

Reviewer #2: No

---

## [Editor Report · Acceptance letter]

31 Aug 2021

PONE-D-21-05273R2 

Acetylcholine esterase inhibitory activity of green synthesized nanosilver by naphthopyrones isolated from marine-derived Aspergillus niger 

Dear Dr. Abdelwahab:

I'm pleased to inform you that your manuscript has been deemed suitable for publication in PLOS ONE. Congratulations! Your manuscript is now with our production department. 

Kind regards, 

on behalf of

Dr. Mohammad Shahid 

Academic Editor

PLOS ONE